# Perils of the PICC: Peripherally Inserted Central Catheter-Associated Complications and Recommendations for Prevention in Clinical Practice—A Narrative Review

**DOI:** 10.3390/healthcare13161993

**Published:** 2025-08-14

**Authors:** Benjamin Kalma, André van Zundert

**Affiliations:** 1Department of Intensive Care, Royal Brisbane and Women’s Hospital, Brisbane, QLD 4006, Australia; benjamin.kalma@health.qld.gov.au; 2Department of Anaesthesia and Perioperative Medicine, Royal Brisbane and Women’s Hospital, Brisbane, QLD 4006, Australia; 3Royal Brisbane Clinical Unit, Faculty of Medicine, The University of Queensland, Brisbane, QLD 4029, Australia

**Keywords:** peripherally inserted central catheters, complication, incidence, prevention, central line-associated bloodstream infection, thrombosis, intensive care, critical care, anaesthesia, review

## Abstract

Background: Peripherally inserted central catheters (PICCs) are becoming an increasingly utilised alternative to traditional central venous access devices. Their uptake, particularly among oncology patients, is due to their growing ease of access, suitability for medium-term use and perceived safety profile. However, PICCs can be a source of severe and life-threatening complications such as central line-associated bloodstream infection (CLABSI), deep vein thrombosis (DVT), pulmonary embolism (PE), malpositioning, dislodgement, and occlusion. Methods: This narrative was constructed from a literature review of the PubMed database, utilising MESH terms for peripherally inserted central catheters, percutaneous central catheters, PICC, and complications. Randomised controlled trials, systematic reviews, and meta-analyses published between 2015 and 2025 were included. Additional articles were obtained through targeted PubMed searches or from references within previous articles. Results: Major periprocedural complications were seen in 1.1% of PICC insertions, CLABSI in 1.4–1.9%, venous thrombosis embolism (including PE) in 2.3–5.9%, and malpositioning in 7.87%. The overall PICC complication incidence was 9.5–38.6%, which is greater than that of centrally inserted central venous access. A higher BMI, diabetes mellitus, chronic renal failure, and malignancy were the most significant predictive factors for PICC-associated complications. Conclusions: PICC complications are common, occurring more frequently than other forms of central venous access, and may lead to significant morbidity and mortality. Appropriate assessment of patient risk factors and optimisation strategies may reduce complication rates.

## 1. Introduction and Brief History of the Peripherally Inserted Central Catheter

Peripherally inserted central catheters (PICCs) are a common component of modern vascular access, utilised widely throughout inpatient and outpatient settings, particularly cancer care services and critical care. They involve the insertion of long, flexible venous catheters through a small peripheral vein to terminate within the distal third of the superior vena cava. In contrast, traditional central venous catheters (CVCs) have a more proximal insertion point into a large, central vein (such as the internal jugular, subclavian, or femoral vein). They terminate within the inferior vena cava (IVC), superior vena cava (SVC), or right atrium (RA), depending on the insertion site and length of line [1].

PICCs have become a viable alternative to standard CVCs for the administration of irritant intravenous medications or total parenteral nutrition (TPN), and for prolonged vascular access or frequent blood sampling. Since their introduction in the 1970s, the procedure has evolved, with revisions to materials, catheter design, insertion technique, and the safety profile of the procedure. Increasing incidence of PICC utilisation is secondary to greater access to insertion (largely due to dedicated vascular access teams), improved patient comfort, modern imaging guidance, and perceived safety compared to standard CVCs [2,3].

PICC lines were first described in 1975 by Hoshal et al. for the administration of TPN in chronically malnourished patients [4]. The PICC aims to provide central access while reducing the risk of life-threatening complications associated with existing central access techniques, such as puncture of large and incompressible arterial vessels, pneumothorax, haemothorax, and cardiac tamponade [5].

Initial PICC designs were single-lumen, silicone elastomer catheters with an external clamp, which were inserted into the superior vena cava after being placed through the basilic or cephalic vein [4]. Currently, most PICCs are third-generation polyurethane, which is a more durable alternative to the original silicone design. Modern clinicians may also choose between antimicrobial and antithrombogenic coatings, valved or valveless PICCs, and single or multi-lumen configurations [6].

This narrative review aims to provide a comprehensive overview of PICC lines and summarise recent evidence for current complication rates and factors predictive of complications. Catheter material selection, anatomical considerations, and comparison between PICCs and other venous access devices will be discussed. We will utilise current evidence to inform the best practices regarding PICC insertion and management, and discuss whether a PICC is the optimal choice for patients in critical care settings.

This review was constructed from a literature review of the PubMed database, utilising MESH terms of “Peripherally inserted central catheter”, “percutaneous central catheter”, and “PICC”, combined with “complication”. It examines peer-reviewed, randomised controlled trials, and systematic reviews and meta-analyses published between 2015 and 2025. Additional articles were obtained through targeted PubMed searches or from references within previous articles.

## 2. Modern PICC Designs

Catheters with antimicrobial impregnation or coating have been developed to reduce central line-associated bloodstream infection (CLABSI), also known as catheter-related bloodstream infection (CRBSI). There is strong evidence supporting the use of antimicrobial coating for traditional CVCs to reduce the incidence of CLABSI [7]. However, there is currently mixed evidence regarding the effectiveness of antimicrobial coating on PICC lines.

A 2016 meta-analysis of eight studies, including 12,879 patients, found a significant reduction in CLABSI with antimicrobial-coated PICCs (relative risk 0.29; 95% CI, 0.10–0.78), the effect of which was more pronounced in patient populations at a greater baseline risk of infection [8]. However, a larger subsequent meta-analysis of twelve studies, with a total of 51,373 patients, failed to replicate this result, demonstrating no statistically significant reduction in CLABSI with antimicrobial-coated lines [9]. Concerningly, in some studies, a significant increase in post-insertion bleeding was observed in patients receiving an antimicrobial-coated line, necessitating the application of a thrombogenic dressing and, in some instances, a pressure dressing [10]. For these reasons, in addition to the cost, antimicrobial-coated PICC lines are not yet used routinely but should be considered amongst high-risk populations (immunocompromised patients, oncology patients, adult burn patients, or infants).

Antithrombogenic coatings have also been trialled to reduce rates of thrombosis, catheter-associated phlebitis and occlusion. These use a variation in hydrophobic polymers, hydrophilic polymers or a biological protein coating to reduce platelet aggregation and activation [11,12]. Table 1 describes the proposed mechanisms underlying antithrombogenic coatings, most of which originate from in vitro models. While in vitro studies of these materials have been promising, there is limited evidence of the efficacy of supposed antithrombogenic PICC coatings in human trials. A large meta-analysis of 42,562 patients saw that antithrombogenic PICC coatings did not result in reduced rates of thrombosis or occlusion [13].

Additional developments within modern PICCs include the incorporation of internal valves, which replace external clamps, thought to reduce thrombotic complications and obstructions [17]. Early randomised trials comparing valved and non-valved PICCs reported positive results, with lower rates of catheter occlusion and infection observed in valved catheters [18]. However, these results have not been replicated. A 2012 prospective study found no statistically significant difference in occlusion rates with valved PICCs [19]. In 2014, Pittiruti et al. reported no statistically significant difference in occlusion, infection, malfunction, or venous thrombosis between the two catheter types in a randomised prospective study [20]. A recent, multi-centre randomised controlled trial of 1098 patients also saw similar rates of infectious and non-infectious complications between valved and non-valved PICCs. However, in this study, using different catheter materials between the study arms limits the effectiveness of direct comparison between valved and non-valved catheters [17]. A 2023 meta-analysis of 2346 patients, the most recent to date, found a decrease in occlusion rates for new PICC designs (including valves); however, it failed to demonstrate statistical significance for any other complication [21]. A summary of recent evidence for antimicrobial, antithrombogenic, and valved PICCs is presented in Table 2.

Modern PICCs include between one and three lumens, allowing different choices depending on clinical need. However, an increasing number of catheter lumens and catheter size have been consistently associated with the incidence of numerous complications, including CLABSI [13,23], catheter-related thrombosis [24], catheter reinsertion [24], and accidental withdrawal and migration [25]. With the growing availability of catheter materials and the number of lumens, clinicians must decide on the appropriate catheter for their patient.

## 3. Indications for PICC Insertion

PICCs should be inserted for an appropriate indication, and the best anatomical site should be selected. There are many clinical scenarios in which PICCs are indicated.

These include the administration of irritant medications, such as chemotherapy and vasoactive agents, and prolonged courses of intravenous medications, including antimicrobials [26,27]. PICCs are also commonly used to infuse hyperosmolar solutions or substances with extreme pH, such as TPN. Patients may also benefit from PICC placement in cases that require frequent venous access, such as those involving multiple transfusions, poor peripheral access, or repeated blood sampling. Other indications for PICC insertion include anatomic abnormalities in the chest and neck that make standard CVC placement difficult, and when required as an alternative to umbilical catheters in neonates [26,27].

Local contraindications to insertion include the presence of skin infections, trauma, burns, or local radiation therapy. PICC placement should be avoided in limbs with pre-existing venous thrombosis or previous mastectomy with lymph node dissection. Systemic contraindications include bacteraemia, severe coagulopathy, and thrombocytopenia due to the increased risk of CLABSI and haematoma formation. Small-calibre vessels (<3 mm) may preclude successful catheter placement and predispose patients to an increased risk of thrombosis. PICC placement is also avoided in patients with chronic kidney disease, who are anticipated to, or currently undergoing, haemodialysis. This is to preserve the venous architecture for future arteriovenous access. Patients with persistent coughing and vomiting may be predisposed to catheter malposition, catheter erosion, or cardiac tamponade [26,27]. Other contraindications include patients who do not provide consent for the procedure or are too agitated to tolerate the insertion.

## 4. Anatomical Considerations

Figure 1 illustrates the vascular anatomy relevant to PICC insertion, while Table 3 compares the anatomical course and characteristics of potential insertion vessels. The most preferred vessel is the basilic vein due to its accessibility, distance from critical structures, and less tortuous path to the central compartment. An illustration of this may also be seen in Figure 1. In neonate populations, head and neck veins, such as temporal veins and posterior auricular veins, can be used for PICC placement. Additionally, saphenous veins in the lower extremity are suitable for PICC line insertion [26].

## 5. Vascular Access Device Selection

When patients are considered for PICC insertion, clinicians must determine if their situation would be more appropriately treated with a different form of device, such as a midline catheter (MC), standard non-tunnelled CVC, or a tunnelled catheter. Table 4 summarises anatomical differences and treatment considerations for selecting such a device.

## 6. Incidence of PICC-Associated Complications

The results of this review have been tabulated, separated by each reported complication (Table A1, Table A2, Table A3, Table A4, Table A5, Table A6, Table A7, Table A8 and Table A9). These have been extracted and summarised in Table 5 to provide a current estimated incidence of PICC-associated complications. A comparison has been made with recent incidences of tunnelled and non-tunnelled CVC complications. As demonstrated, PICC complications are common, occurring in as many as 38.6% of inserted devices, a substantially greater rate than CVCs [17,34,35].

Although major periprocedural complications such as pneumothorax, arterial injury, and bleeding appear to occur at lower rates for PICC lines, PICCs are malpositioned in up to 8.87% of cases. Catheter occlusion and dislodgement are also common, with reported incidences as high as 33.6% and 8%, respectively. Finally, life-threatening delayed complications such as CLABSI and thrombosis (including PE) appear to occur at a greater rate than standard CVCs.

The most significant risk factors for any PICC-associated complication have been identified as higher BMI, diabetes mellitus, chronic renal failure, and malignancy [25,35]. Major complications, defined as infectious or non-infectious complications requiring device removal, occurred in 5.2% to 9.9% of devices [17,46].

### 6.1. Central Line-Associated Blood Stream Infection (CLABSI)

Infections are among the most serious catheter-associated complications and can include cellulitis, abscesses, and systemic bloodstream infections. Bacteraemia may be complicated by septic shock, infective endocarditis, septic emboli, and mortality [50]. The most common causative organisms of catheter-related infections are coagulase-negative Staphylococci, Staphylococcus aureus, Enterococci, and Candida [27]. CLABSI is especially detrimental in an immunocompromised population, such as those with active malignancy or receiving cancer treatment, with an estimated mortality between 31 and 36% [30].

For this reason, significant effort has been made to identify the incidence, risk factors, and prognostic factors for catheter-related infections in haematology, oncology, and critical care patients. Over the past two decades, three large meta-analyses have examined the incidence of CLABSI in the PICC population. Most studies require a positive blood culture to diagnose CLABSI [2]. In 2006, Maki et al. demonstrated an incidence of 2.4% or 2.1 per 1000 catheter days [51]. Subsequently, Chopra et al. reported similar rates of between 1.0 and 2.1 per 1000 catheter days in all patient populations and a higher incidence of 1.81–7.71 per 1000 catheter days among oncology patients [30]. This result is supported by the most recent study, a 2025 Australian multi-centre randomised controlled trial of 1098 patients, which reported that CLABSI affected between 1.4% and 1.9% of participants [17]. Other recent prospective studies have reported similar results, with rates of 1.3% and 1.6% [35,46].

Among other access devices, a 2020 analysis comparing PICCs to standard CVCs reported a statistically significantly higher incidence of CLABSI among patients with PICCs than those with standard CVCs (as described in Table 5) [2]. Direct comparison analyses of PICCs and midline catheters have yielded varied results, with Urtecho et al. reporting lower rates of catheter-related bloodstream infections in midline catheter populations [37]. Studies by Lai et al. and Lu et al. have also reported no significant differences in the rate of bloodstream infections [31,52].

Risk factors for CLABSI are well-documented in the literature and are important for clinicians to consider when selecting patients for PICC line insertion. The risk for PICC-associated CLABSI increases significantly in patients with severe or critical illness, immunosuppression, or malignancy (the greatest and most significant risk being aggressive haematological malignancies and the lowest for oesophageal and colorectal cancers) [30]. Other comorbidities such as obesity (BMI > 30 kg/m^2^) and diabetes mellitus were also associated with CLABSI development [25]. Catheter and patient factors, including an increased number of PICC lumens, longer catheter placement duration, and a greater number of previous CVC insertions, also significantly increased the risk of CLABSI [30,53]. Tunnelled PICCs have consistently seen lower rates of CLABSI compared to standard PICC insertion across meta-analyses [50,54].

Other interventions, such as newly developed dressing securement devices, have not shown significant differences in overall complication or infection rate compared to standard dressings [25].

### 6.2. Catheter-Associated Thrombosis

Catheter-associated thrombosis accounts for significant morbidity and mortality amongst PICC complications due to its high incidence and the risk of clot propagation and pulmonary embolism [30]. Most studies assessed patients for deep vein thrombosis (DVT) only when symptomatic, and the most common method of diagnosis was ultrasonography. Few studies have used venography alone to diagnose DVT [55].

The cohort with the significant risk for thrombosis is patients in the intensive care unit, with significantly greater rates of PICC-related DVT in ICU patients (13.91%) compared to oncology patients (6.67%) and general ward patients (3.44%) in a 2013 meta-analysis [55].

Several other meta-analyses have investigated the incidence of thromboembolism in PICC populations, yielding considerable variation in results. A 2003 review reported an incidence of DVT between 0.3% and 28.3% [56]. More recent meta-analyses in 2012 and 2019 yielded similar results, with catheter-associated thrombosis occurring at rates of 2.0–5.5% and 2.4%, respectively, in general populations, and at a higher rate in populations with cancer (3.4–7.8% and 5.9%, respectively) [30,57]. Another recent meta-analysis, published in 2021 by Schears et al., reported a similar absolute risk of 2.3% for venous thromboembolism after PICC insertion [48]. Other reviewed prospective studies reported PICC-associated DVT incidences ranging from 1.5% to 11% between 2017 and 2025 [34,35,44,46,47].

Further analyses have been conducted to compare PICC-associated thrombosis with other vascular access devices, such as standard CVCs, midlines, or tunnelled devices. Chopra et al. observed a statistically significant increase in the occurrence of thrombosis in PICC lines compared to standard CVCs [55], a finding supported by subsequent results from Mavrovounis et al. in ICU patients and oncology patients [2]. No difference between rates of catheter-associated thrombosis was seen between PICCs and midline catheters for adults and children in two large meta-analyses [31,37]. Multiple studies have demonstrated a lower incidence of catheter-associated thrombosis in tunnelled PICCs compared to standard non-tunnelled lines. A recent meta-analysis of five RCTs with a total of 1238 patients saw significantly lower rates of thrombosis in tunnelled PICCs [54]. These results were consistent with a previous meta-analysis, which also demonstrated similar findings [50]. This is theorised to be due to the catheter’s tunnelling, allowing access to the vessel at a point more proximally, thereby increasing the ratio of catheter to vein diameter [58]. In 2023, Li et al. investigated the optimal length for a subcutaneously tunnelled PICC line and observed a significant decrease in some PICC-related complications for patients with a tunnelled length greater than four centimetres [59]. These complications were wound oozing, catheter dislodgement, and unplanned catheter removal. However, this study did not demonstrate a significant difference between tunnelled and standard PICCs in terms of thrombosis, infection, or catheter occlusion.

Among the examined studies, patient risk factors for thrombosis include diabetes mellitus, active malignant disease, current chemotherapy, chronic renal failure, and a history of prior venous thromboembolism. Clinical risk factors include PICC placement in a paretic arm, mannitol use, larger catheter size, and prolonged surgical operation (>1 h) while the PICC is in place [25,44,60,61,62].

The peripheral vein insertion site has also been assessed for its effect on thrombosis. A 2018 prospective study examined the differences in complications between brachial and non-brachial vein PICC access [44]. This study saw no difference in the incidence of thrombosis between the two subgroups; however, it did demonstrate a significantly longer procedure time and a greater number of obstacles encountered with non-brachial access (without use of ultrasound).

Novel work has investigated serum biomarkers that may be predictive of catheter-related thrombosis. Elevated serum D-Dimer and platelets were associated with the presence of DVT. Other examined biomarkers included APTT, fibrinogen, FDP, glucose, haemoglobin, glycated haemoglobin, INR, prothrombin time, prothrombin fragment 1.2, the thrombin-antithrombin complex, and WBC; however, none had a positive association with DVT development [63].

### 6.3. Catheter Occlusion

Catheter occlusion is a significant failure, with potential risk of life-threatening loss of venous access in critical care settings. Secondary effects may include economic costs and patient distress from catheter reinsertion [26,46]. Occlusion may be intraluminal secondary to coagulated blood or precipitated infusion products, or extraluminal secondary to fibrin sheath formation, thrombosis, or incorrect catheter positioning (such as kinking or abutment against the vessel wall) [26]. The reported rates of catheter occlusion show considerable variance; most studies report an incidence ranging from 1.9% to 7% [35,44,46]. Some others have reported results ranging from 12% to 33.6% [17,25]. These variations are likely affected by the patient populations, the catheter materials used, and institutional protocols for PICC management.

### 6.4. Catheter Dislodgement

Catheter dislodgement occurs at higher rates in inpatient populations and patients with non-tunnelled catheters [25,50], with a total reported frequency ranging from 2.3% to 8% in recent years [25,35,44]. Catheters with a greater number of lumens were also at a higher risk of dislodgement [25].

### 6.5. Phlebitis

Phlebitis may be caused mechanically by the catheter or through chemical irritation by medications [27]. Mechanical phlebitis has been reported at a low rate in recent trials; approximately 1.3% of PICCs were affected [39]. Non-significant differences were seen between rates of phlebitis in PICCs and midline catheters, although tunnelled PICCs saw a significantly decreased phlebitis risk [50].

### 6.6. Medical Adhesive-Related Skin Injury (MARSI)

MARSI may include mechanical skin injury, contact dermatitis, folliculitis, and moisture-associated skin damage. Frequent changes of PICC dressings are crucial for maintaining asepsis and preventing contamination of the catheter skin entry point. However, this action predisposes a patient’s skin to damage. A 2025 meta-analysis reported an incidence of 22% among oncology patients with PICCs [49]. Key risk factors identified for the development of MARSI were advanced age, a higher BMI, pre-existing skin conditions (such as eczema or dermatitis), and the use of transparent film dressings. There were no statistically significant differences between rates of MARSI in tunnelled and standard PICCs [54]. Educating healthcare providers regarding MARSI risk assessment, prevention, and management significantly decreased the incidence in one small prospective study [64].

## 7. Peri-Procedural Complications

A range of complications may occur peri-procedurally during PICC insertion. Historically, this risk has been considered lower than that of standard CVCs [2,3]. Notably, the risk of serious complications, such as large artery puncture, pneumothorax, haemothorax, and cardiac tamponade, is reduced compared to traditional CVCs, as presented in Table 5.

### 7.1. Malpositioning

A common complication is catheter malpositioning, where the PICC tip does not reside in the SVC. Primary malposition occurs on insertion, while secondary malposition is the result of spontaneous migration at some point after insertion. Migration carries significant concern due to the possibility that it may cause tamponade via erosion of the catheter through the right atrium or ventricle [27]. A 2013 retrospective study of 3012 patients observed malpositioning in 7.87% of cases [40]. The overall most frequent site of malpositioning was the internal jugular vein (3.23%), followed by the axillary vein (2.16%). The chosen insertion site appeared to affect the most likely location to be malpositioned. PICCs inserted into the cephalic vein were most commonly malpositioned in the axillary and basilic veins. Basilic vein PICCs were most commonly malpositioned in the internal jugular vein. The azygos vein is another site of reported malposition [65]. Actions such as slowly threading the catheter and patient positioning (lateral rotation and flexion of the neck) may reduce the likelihood of malpositioning [40].

Malpositioning is also an important consideration of centrally inserted catheters, where it is defined as the catheter tip not terminating in the region between the distal SVC and cava–atrial junction [41]. CVC malpositioning may be described as intra-cava and extra-cava, examples of which are displayed in Table 6. Catheter malposition may be due to inaccurate and deep cannulation of the vessel, anatomical variations, blocked IVC or SVC, elevated central venous pressure, or patient positioning [41]. Depending on the location of the catheter, management may vary from cautious removal or radiologically guided repositioning to surgical intervention. The incidence of CVC malpositioning is estimated to occur in 3.3–5.01% of cases [41,42].

### 7.2. Bleeding/Haematoma Formation

Recent randomised controlled trials have shown the incidence of major bleeding or haematoma to range from 1.3% to 3% among standard PICCs [25,44]. Multiple meta-analyses comparing standard and tunnelled PICCs saw a decrease in post-procedure wound oozing. However, there are no significant differences in rates of major bleeding or haematoma formation [50,54].

### 7.3. Nerve Injury

Another complication of PICC insertion is nerve injury, which, in one study, was reported as an incidence of 0.15% [44]. There was no difference in the incidence of nerve injury for standard or tunnelled PICCs [54]. There are numerous case reports of nerve injury syndromes associated with PICC placement, with the median interosseous nerves being the most commonly affected [66,67,68].

### 7.4. Rare Complications

There are case reports of severe and rare complications secondary to PICC insertion. Such complications include air embolism, arrhythmias (usually secondary to deep guidewire insertion), and accidental guidewire retention [69,70,71].

Arterial puncture is a well-documented complication of standard CVC insertion and carries significant associated morbidity [5,72]. None of the analyses reviewed for this article directly assessed the incidence of accidental arterial puncture during PICC insertion. This may have been reflected under rates of general procedure success or total complication rates, although they did not make any distinction of this outcome. Both arterial puncture and arterial PICC line placement have been reported to occur within the literature. Known complications of arterial puncture include brachial arteriovenous fistula, pseudoaneurysm formation, and the potential for distal arterial flow compromise [73,74]. Inadvertent arterial PICC placement is theorised to be more common in infants and neonates due to lower blood pressure and smaller arterial diameters, meaning radiographic signs typically used for differentiation between venous and arterial vessels are not present on ultrasonographic examination. Other rare PICC-associated complications reported in children have been pericardial effusion, pleural effusion, and pneumothorax [75,76,77].

Other acute complications with a recently reported incidence include skin allergy and liquid extravasation, seen in a 2020 randomised trial of 2250 patients to occur at rates of 4.6% and 2.9%, respectively [34].

## 8. Paediatric Populations

While this review has predominantly focused on PICC-associated complications within the adult population, paediatric populations also see significant rates of complications necessitating catheter removal, occurring as frequently as 41% of cases [78]. Complications seen in children are similar to those in adults, including mechanical (such as DVT, extravasation, occlusion, displacement, injuries to surrounding structures, pneumothorax, and cardiac tamponade) and infections (such as CLABSI).

Several risk factors for PICC-associated complications have been identified that are specific to the paediatric population. These factors include younger age (particularly < 1 year), paediatric intensive care admission, double-lumen catheters, more frequent catheter access, catheter dwell time, non-central catheter tip location, and catheter insertion site [79,80]. Clinicians should consider these factors when evaluating PICC insertion in paediatric populations and closely monitor for both acute and delayed complications.

## 9. Recommendations to Reduce PICC-Associated Complications

Clinicians must critically assess a patient’s history and treatment requirements before inserting a PICC line, considering the risk factors previously discussed in this review.

The routine use of ultrasound guidance for insertion, imaging confirmation of tip positioning in the SVC and measurement of selected vein diameter before insertion are accessible strategies to reduce PICC-associated complications, predominantly venous thromboembolism (VTE) risk [29]. A summary of management strategies and a suggested risk assessment are provided in Table 7.

Other successful implementations include the growing practice of dedicated vascular access teams within hospitals to assist in inserting and managing PICC lines. Self-efficacy and patient education programmes have also been shown to reduce overall complication rates, including decreased infection rates, catheter occlusions, and accidental catheter withdrawals [53].

### 9.1. Prophylactic Anticoagulation

There is currently limited evidence supporting routine prophylactic anticoagulation for prevention of PICC-associated thrombosis, and thus modern guidelines do not suggest this practice [81]. In retrospective studies, anticoagulated patients have significantly decreased rates of PICC line-associated venous thrombosis [82]. However, most of the evidence has examined VTE prophylaxis for all types of central venous catheters, rather than specifically PICCs. Most recently, a 2018 Cochrane meta-analysis described moderate certainty evidence that low molecular weight heparin (LMWH) reduces catheter-related VTE compared to no LMWH, although it did not find a conclusive effect on mortality [83]. Vitamin K antagonists were also not conclusively seen to reduce catheter-related VTE or mortality.

Subsequently, a 2021 meta-analysis of 12 clinical trials by Li et al. found a lower incidence of VTE, at 7.6% among patients receiving VTE prophylaxis compared to 13.0% in those not receiving it (OR 0.51, 95% CI 0.32 to 0.82, *p* < 0.01) [84]. They included trials using LMWH, vitamin K antagonists, and direct oral anticoagulants, collectively demonstrating similar rates of major bleeding and higher rates of minor bleeding.

Although current recommendations are not for routine use of prophylactic anticoagulation, it is certainly reasonable to consider prophylaxis in high-risk population groups (such as those with previous VTE or malignancy).

### 9.2. Monitoring of Complications

Immediate complications may become clinically evident in minutes to hours post-procedure, and thus close monitoring is recommended for this period [26]. When assessing delayed complications, a 2017 prospective cohort study saw an average time-to-complication of 8 and 16 days for accidental withdrawal and occlusion, respectively [85]. A retrospective study reported the average time-to-complication to be 2.23 days for phlebitis, 28.9 days for thrombosis, 76.48 days for occlusion, 114.26 days for infection, and 163.75 days for catheter migration [86]. Finally, a 2017 retrospective analysis found a time-to-complication of 5 days for thrombosis and 16 days for infection [87]. These differences are likely secondary to variations among patient comorbidities, inpatient status, and differing insertion techniques; however, they highlight that delayed complications may occur as early as the first week and must continue to be vigilantly observed for months after line insertion.

Although there is less data available for paediatric populations, delayed complications appear to occur within similar time frames in children as in adults. Two studies have reported median catheter dwell times prior to line removal of 13 days and 17.7 days, respectively [80,88].

## 10. Limitations of Literature and Future Studies

Many of the large studies reviewed in this article did not report on significant confounding variables. In one analysis, 58% of studies did not report whether the patient was on chemical VTE prophylaxis, and 31% of articles did not describe whether the position of the PICC tip was ascertained after insertion [55]. These are known risk factors for PICC-associated complications and likely confound results.

There was significant variability in patient populations across the reviewed articles, with few studies examining patients solely in oncology or intensive care settings. The heterogeneity between these populations results in significant variation among the reported outcomes and limits the generalisability of the results to other populations. Additionally, most studies included within the systematic reviews and meta-analyses were retrospective or observational studies, which are prone to bias.

The economic effect of PICC-associated complications is likely underappreciated. It would be beneficial for future investigations to include an analysis of the cost-effectiveness of PICCs and CVCs. This would involve examining upfront costs (catheter insertion, maintenance, and removal) and the financial burden of managing complications such as CLABSI, VTE, and mechanical failures. Such analysis would guide decision-making for patient and catheter selection, policy creation, and resource allocation.

As evidenced by the results of this review, there are a few high-quality prospective trials that directly compare the outcomes of traditional CVCs and PICCs. Further trials among populations of interest (particularly intensive care) or niche populations such as paediatrics are required to conclude adequately. Furthermore, few studies in this review reported mortality rates or long-term functional outcomes, which are an important consideration for high-risk delayed complications such as CLABSI or catheter-associated thrombosis.

## 11. Conclusions

Although historically viewed as a safer alternative to central access, this review has demonstrated that PICC-associated complications are common, with total complication rates up to 38.6%. Major procedural complications, such as pneumothorax, haemothorax, cardiac tamponade, and arterial injury, appear to be lower with traditional CVCs. However, PICC complications may still carry significant morbidity and mortality. Serious delayed complications such as CLABSI or thrombosis have consistently been seen to occur at greater rates with PICCs than other forms of central access. Although they have a role in medium-term access, PICCs have a significantly higher risk of complications among critical care and immunosuppressed populations. For these reasons, the treating clinician must assess a patient’s clinical background and carefully tailor their selection for venous access. In critical care settings, a standard CVC or a tunnelled CVC may be a more appropriate choice for many patients.

## Figures and Tables

**Figure 1 healthcare-13-01993-f001:**
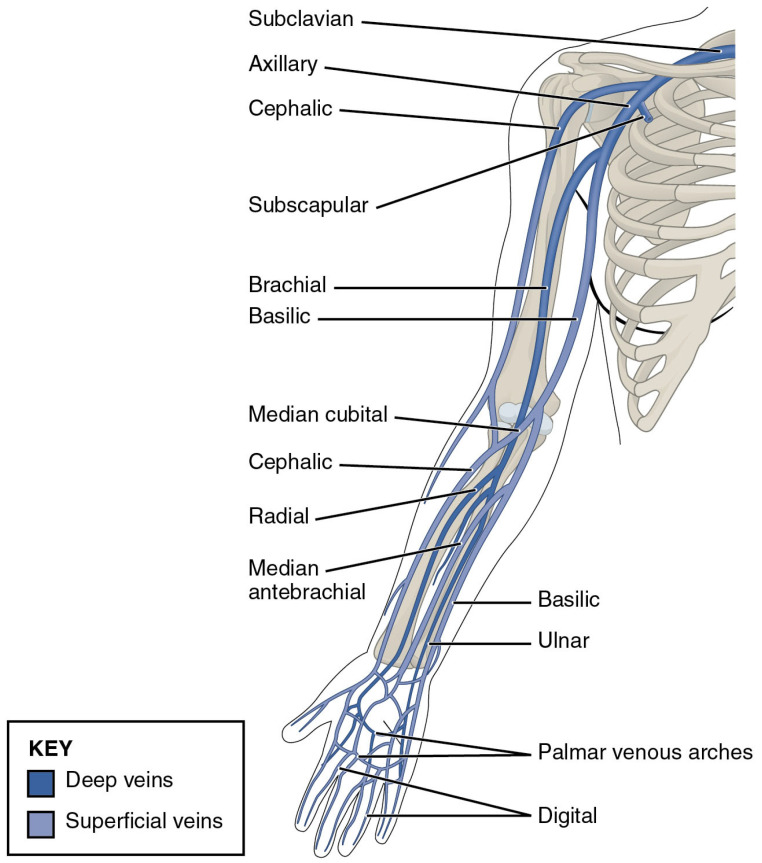
Peripheral venous sites of PICC line insertion, i.e., basilic vein, brachial vein, and cephalic vein. Illustration from Anatomy & Physiology, Connexions Web site. Accessed 5 August 2025. http://cnx.org/content/col11496/1.6—with permission [28].

**Table 1 healthcare-13-01993-t001:** Proposed mechanisms of different antithrombogenic materials [6,12,14,15,16].

Coating	Proposed Mechanism	Example Material/Brand
Hydrophobic polymer	Reduces the contact area between the catheter surface and proteins or platelets	BioFlo^®^ PICC (Angiodynamics, Latham, NY, USA)
Hydrophilic polymer	Readily forms hydrogen bonds with surrounding water molecules, creating a highly hydrated surface layer. This acts as a physical barrier that inhibits the adhesion of proteins and bacteria.	Polyethylene Glycol (PEG), HydroPICC^®^ (Access Vascular, Billerica, MA, USA)
Biological protein	Utilises proteins with antithrombogenic properties to resist protein adsorption and platelet adhesion on surfaces	Albumin coating
Heparin-bonded	Interacts with plasma protein antithrombin, reducing fibronectin deposition	Hygea^®^ (Jmedtech, Singapore)
Combined hydrophilic and hydrophobic	As above	SEC-1^®^ (Toyobo, Osaka, Japan)—composed of PEG and a hydrophobic alkyl group

Abbreviations: Peripherally inserted central catheter (PICC); polyethylene glycol (PEG).

**Table 2 healthcare-13-01993-t002:** Summary of recent evidence for PICC-line coatings and valves [9,22].

Intervention	Subcategories	Study	Patients	Evidence
Antimicrobial coating	Antiseptic—CHG	MA	12,879	Statistically significant reduction in CLABSI, especially in populations at greater baseline risk of infection [9] No statistically significant reduction in CLABSI, increased risk of bleeding complications with antimicrobial coating [22]
	Antibiotic—MNC and RMP	MA	51,373	
	Antifungal—MCZ			
Antithrombogenic coating	Hydrophobic polymers	MA	42,562	No effect on rates of thrombosis or occlusion [13]
	Hydrophilic polymers			
	Biological proteins			
	Heparin-bonded			
Valved PICC	Valved vs. non-valved PICC	RCT RCT RCT RCT MA	362 120 182 1098 2346	Valved clamps reduced occlusion and infection rates [18] Presence/valve type did not significantly influence occlusion rates [19] No significant difference in complications [20] Similar complication rates (however, variation in materials) [17] New PICC design (including valves) reduces occlusion but no other complications [21]

Abbreviations: chlorhexidine (CHG); minocycline (MNC); rifampin (RMP); miconazole (MCZ); central line-associated bloodstream infection (CLABSI); meta-analysis (MA); randomised controlled trial (RCT); peripherally inserted central catheter (PICC). Thrombosis includes deep venous thrombosis and pulmonary embolism.

**Table 3 healthcare-13-01993-t003:** Anatomical choices for PICC insertion in adults [27].

Vein	Anatomical Location	Advantages and Disadvantages	Comment
Basilic *	Arises from the dorsal vein network of the hand. Ascends the medial aspect of the upper limb. Moves from superficial to deep at the level of the teres major. Combines with the brachial vein to form the axillary vein.	The least torturous path from insertion point to the SVC. Accessing it may be more challenging and complex without ultrasound guidance.	Preferred first-line access site for PICC insertion.
Brachial *	Formed by the union of the ulnar and radial veins at the elbow. Often co-located with the brachial artery, runs deep in the arm, medial to the humerus. Combines with the basilic vein to form the axillary vein.	Good vessel calibre. Less tortuous than the cephalic vein. Increased risk of accidental arterial puncture and nerve injury (proximity to the brachial artery and median nerve).	Suitable second-line access, particularly with USS guidance.
Cephalic *	Arises from the dorsal vein network of the hand. Ascends the anterolateral aspect of the upper limb. Runs along the lateral aspect of the biceps and enters the axilla region via the clavipectoral triangle, where it empties into the axillary vein.	Easier to access without USS, less risk of nerve injury. Smaller vessel calibre. A more tortuous path leads to challenging catheter advancement.	Less commonly preferred due to tortuosity and smaller size.

* Insertion should occur in the proximal arm above the elbow due to a lower risk of thrombosis, phlebitis, and infection [27,29]. Abbreviations: superior vena cava (SVC); ultrasound scan (USS).

**Table 4 healthcare-13-01993-t004:** Comparison of different vascular access devices [27,30].

Device	MC	CVC	PICC	Tunnelled CVC
Catheter insertion point	Peripheral arm vein	Jugular, subclavian or femoral vein	Peripheral arm vein	Jugular, subclavian or femoral vein
Tip location	Axillary or subclavian vein (does not reach central compartment)	Cavo-atrial junction (except femoral catheters)	Distal superior vena cava	Cavo-atrial junction (except femoral catheters)
Safe duration	Several weeks	<2 weeks	<6 months	Months to years
Substances able to be safely administered	Fluids Blood products Non-vesicant medications Vasopressors	Fluids Blood products Non-vesicant medications Vasopressors Vesicant medications TPN	Fluids Blood products Non-vesicant medications Vasopressors Vesicant medications TPN	Fluids Blood products Non-vesicant medications Vasopressors Vesicant medications TPN
Overall complication incidence	18.85% [31]	5.9–7% [32,33]	9.5–38.6% [17,34,35]	22.1% [36]
CLABSI incidence	0.33% [37]	0.48% [5]	1.4–1.9% [17]	2.81 cases per 1000 days [38]
Other advantages	Lower infection risk than CVC Fewer insertion complications Lower risk of phlebitis and venous occlusion compared to CVC	Haemodynamic monitoring Can be placed at the bedside (in ICU)	Lower insertion complication rate than CVC Suitable for outpatient use Can be placed at the bedside on the ward by trained nursing staff	Lower rates of catheter-associated thrombosis, CLABSI and phlebitis
Limitations	Not suitable for vesicant medications or TPN Some studies have shown higher rates of **any** complication compared to PICCs (RR 1.95) [31]	Insertion complications with significant morbidity/mortality Femoral site greater risk of infection Comparatively short-term access	Greater thrombosis and CLABSI risk compared to CVC Requires regular maintenance and dressing changes	Surgical insertion required Requires regular maintenance and dressing changes

Abbreviations: midline catheter (MC); central venous catheter (CVC); intensive care unit (ICU); total parenteral nutrition (TPN). Vesicant medications include chemotherapies, inotropes, certain vasopressors, and high-concentration electrolyte solutions.

**Table 5 healthcare-13-01993-t005:** Incidence of PICC and CVC-associated complications.

Complications	PICC	Non-Tunnelled CVC	Tunnelled CVCs (Including Ports)
**Overall**	9.5–38.6% [17,34,35]	5.9–7% [32,33]	22.1% [36]
**Mechanical**			
Overall major periprocedural	1.1% [36]	0.7–2.1% [5]	2.7% [36]
Pneumothorax	0% [36]	0.44% [5]	0.6% [36]
Inadvertent arterial puncture	NA	1.62% [5]	NA
Inadvertent arterial cannulation	NA	0.28% [5]	0.1–0.8% [39]
Malpositioning	7.87% [40]	3.3–5.01% [41,42]	0–4% [43]
Nerve injury	0.15% [44]	NA	NA
**Bleeding**			
Bleeding or haematoma	1.3–3.0% [25,44]	4.6% [45]	up to 8.0% [39]
Cardiac tamponade	NA	0.01–0.3% [5]	NA
**Delayed mechanical**			
Thrombosis	2.3–11.0% [34,35,44,46,47,48]	0.27–3.0% [5]	3.0% [36]
Occlusion	1.9–33.6% [17,25,35,44,46]	NA	NA
Dislodgement	2.3–8% [25,35,44]	NA	NA
Phlebitis	1.3% [34]	NA	NA
MARSI	22% [49]	NA	NA
**Infectious**			
CLABSI	1.4–1.9% [17]	0.48% [5]	2.81 cases per 1000 days [38]

Abbreviations: medical adhesive-related skin injury (MARSI); data not available in included articles (NA).

**Table 6 healthcare-13-01993-t006:** Summary of potential locations of intra-cava and extra-cava catheter misplacement during CVC insertion [41].

Intra-Cava Misplacement	Extra-Cava Misplacement
Carotid artery Azygos vein Persistent left-sided SVC (anatomical variation) Internal mammary vein Vertebral vein Other veins	Extradural space Pericardium Pleural space Mediastinum Thoracic duct

**Table 7 healthcare-13-01993-t007:** Recommended risk assessment and management strategies in patients considered for PICC insertion to prevent PICC-associated complications [29,30]. Abbreviations: venous thromboembolism (VTE); electrocardiogram (ECG).

Risk Assessment	Recommendation
Patient factors	Review of patients’ history for malignancy or previous VTE Awareness of high-risk comorbidities (renal failure, obesity, diabetes mellitus) Consider an alternate line in patients with current critical illness or active bacteraemia Review clinical indication and line choice if multiple previous central line insertions
Expected treatment course	Selection of the least invasive device required to administer the required treatment safely Avoid insertion of PICC if the patient has an upcoming surgical procedure > 1 h in duration due to elevated thrombosis risk Remove PICC at the earliest opportunity or if concerns for major complications
Procedural factors	Use of ultrasonography for insertion Assessment of selected vein for diameter >3 mm at insertion site Maintenance of strict asepsis during procedure Use of the minimum number of lumens required for treatment Consideration of antimicrobial-coated PICC in high-risk populations
Post-procedure assessment	Imaging or ECG guided confirmation of catheter tip position in SVC Consideration of chemical VTE prophylaxis if not already indicated Access to the device only by appropriately trained staff Frequent monitoring for acute and delayed complications Patient and healthcare staff education regarding safe PICC management

## Data Availability

The original contributions presented in this study are included in the article. Further inquiries can be directed to the corresponding author.

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
