# Peer review of "Perils of the PICC: Peripherally Inserted Central Catheter-Associated Complications and Recommendations for Prevention in Clinical Practice—A Narrative Review"

_healthcare, 2025, doi:10.3390/healthcare13161993_

Round 1

Reviewer 1 Report

Comments and Suggestions for Authors

Thank you for the opportunity to review this paper.

The aim of this narrative review is to provide an overview of the PICC lines - related complications,  to identify the patient and catheter- related risk factors for complications and  to define recommandations for prevention in clinical practice (see title).

This paper is based on an extended bibliography- big data, providing an update on the PICC lines use.

Minor comments:

  • In order to in order to increase the impact of this paper on the clinical practice, please extend the discussion at Paragraph 8 , about the recommandations  included in Table 7:

      - The patient history for risk stratification (i.e. previous DVT, multiple previous central line insertion) : the evaluation is the same for adult and for pediatric population? Add a comment about the probability of more than one (concomitent) complications and the risk for the same complication or an additional risk for more complications. Add a comment about the preventive therapy in patients with history of complications (i.e anticoagulant therapy for prior PE, prior DVT, or malignancies associated with hypercoagulability states- high risk for thrombosis- therapy with prophylactic dose of anticoagulant drug ). Please specify if there are differences in recommandations for adult vs. pediatric patients.

- Please extend the discussion about the post-procedure monitoring for early and delayed complications: are any recommandations about the time line of follow-up related to every complication, based on the  patient profile (indications, history, comorbidity), the procedure factors) and age-based recommandations for the monitoring (adult vs. pediatric patients)

Author Response

Response to Reviewer 1 Comments

1. Summary

Thank you very much for taking the time to review this manuscript. Please find detailed responses below and the corresponding revisions/corrections highlighted/in track changes in the re-submitted files.

3. Point-by-point response to Comments and Suggestions for Authors

Comment 1: The patient history for risk stratification (i.e. previous DVT, multiple previous central line insertions): Is the evaluation the same for the adult and pediatric populations?
Add a comment about the probability of more than one (concomitant) complications and the risk for the same complication or an additional risk for more complications.
Add a comment about the preventive therapy in patients with history of complications (i.e anticoagulant therapy for prior PE, prior DVT, or malignancies associated with hypercoagulability states- high risk for thrombosis- therapy with prophylactic dose of anticoagulant drug ).
Please specify if there are differences in recommendations for adult vs. pediatric patients.

Response 1:

Thank you kindly for your response. We agree with your points that these would be of great value and have added a significant portion to sections 8 and 9 to include your suggestions. We are certain that the addition of further risk factors would increase the risk of complications; however, none of the articles we reviewed provided results comparing the incidence of complications with single versus multiple risk factors. This may be an excellent idea for future investigations.

Changes made to the text:
Lines 412-423:

8. Paediatric Populations

While this review has predominantly focused on PICC-associated complications within the adult population, paediatric populations also see significant rates of complications necessitating catheter removal, occurring as frequently as 41% of cases [77]. Complications seen in children are similar to those in adults, including mechanical (such as DVT, extravasation, occlusion, displacement, injuries to surrounding structures, pneumothorax, and cardiac tamponade) and infections (such as CLABSI).

Several risk factors for PICC-associated complications have been identified that are specific to the paediatric population. These factors include younger age (particularly < 1 year), paediatric intensive care admission, double-lumen catheters, more frequent catheter access, catheter dwell time, non-central catheter tip location, and catheter insertion site [78, 79]. Clinicians should pay attention to these factors when considering PICC insertion among paediatric populations and observe closely for acute and delayed complications.

Lines 438-454:

9.1 Prophylactic anticoagulation

There is currently limited evidence supporting routine prophylactic anticoagulation for prevention of PICC-associated thrombosis and thus modern guidelines do not suggest this practice [80]. In retrospective studies, anticoagulated patients have significantly decreased rates of PICC line-associated venous thrombosis [81]. However, most of the evidence has examined VTE prophylaxis for all types of central venous catheters, rather than specifically PICCs. Most recently, a 2018 Cochrane meta-analysis described moderate certainty evidence that low molecular weight heparin (LMWH) reduces catheter-related VTE compared to no LMWH, although without conclusive effect on mortality [82]. Vitamin K antagonists were also not conclusively seen to reduce catheter-related VTE or mortality.

Subsequently, a 2021 meta-analysis of 12 clinical trials by Li et al. found a lower incidence of VTE at 7.6% among patients receiving VTE prophylaxis compared to 13.0% in those not receiving it (OR 0.51, 95% CI 0.32 to 0.82, p < 0.01) [83]. They included trials using LMWH, vitamin K antagonists, and direct oral anticoagulants, collectively demonstrating similar rates of major bleeding and higher rates of minor bleeding.

Although current recommendations are not for routine use of prophylactic anticoagulation, it is certainly reasonable to consider prophylaxis in high-risk population groups (such as those with previous VTE or malignancy).

Comments 2: Please extend the discussion about the post-procedure monitoring for early and delayed complications: are any recommendations about the timeline of follow-up related to every complication, based on the patient profile (indications, history, comorbidity), the procedure factors) and age-based recommendations for the monitoring (adult vs. pediatric patients).

Response 2: Thank you very much for your comments. We have included a discussion regarding the timeframe during which most complications are typically seen post-insertion to guide monitoring.

Changes made to the text:

Lines 456-471:

9.2 Monitoring of complications

Immediate complications may become clinically evident in minutes to hours post-procedure, and thus close monitoring is recommended for this period [26]. When assessing delayed complications, a 2017 prospective cohort study saw an average time-to-complications of 8 and 16 days for accidental withdrawal and occlusion, respectively [84]. A retrospective study reported the average time-to-complication to be 2.23 days for phlebitis, 28.9 days for thrombosis, 76.48 days for occlusion, 114.26 days for infection, and 163.75 days for catheter migration [85]. Finally, a 2017 retrospective analysis found a time-to-complication of 5 days for thrombosis and 16 days for infection [86]. These differences are likely secondary to variations among patient comorbidities, inpatient status and differing insertion techniques; however, they highlight that delayed complications may occur as early as the first week and must continue to be vigilantly observed for months after line insertion.

Although there is less data available for paediatric populations, delayed complications appear to occur within similar time frames in children as adults.

Two studies have reported median catheter dwell time prior to line removal of 13 days and 17.7 days respectively [79,87]. 

Reviewer 2 Report

Comments and Suggestions for Authors

This manuscript presents a comprehensive and well-structured narrative review on peripherally inserted central catheters (PICCs), their associated complications, and current evidence regarding preventive strategies. The topic is highly relevant and timely, particularly given the increasing use of PICCs across a wide range of clinical settings, including critical care and oncology. The paper is clearly written and generally well supported by the literature.

Overall, this is a very interesting and informative review. However, I would like to raise several comments that, if addressed, could significantly strengthen the manuscript:

1. The authors state in Table 1 (lines 104–105) that both hydrophobic and hydrophilic polymer coatings have antithrombogenic properties. While this may be true from a theoretical or in vitro perspective, it is surprising. The conclusion that both types of coating are antithrombogenic should be further contextualized.

2. The authors describe in Table 4 (lines 178–181) that only CVCs may be inserted into femoral veins. This is an oversimplification. Tunnelled CVCs can also be placed via femoral access, particularly in patients where upper body access is contraindicated. Clarifying this point would improve the objectivity and completeness of the discussion on vascular access selection. It would also help avoid under-representing the versatility of tunnelled catheters in clinical practice.

3. It is unclear why inadvertent arterial puncture is not listed as a potential complication of PICC insertion (Table 5). The basilic vein runs in close proximity to the brachial artery, and arterial puncture has been described in the literature as a complication of PICC insertion. This omission may create an impression of differential risk that is not fully supported by the evidence.

Similarly, the table reports inadvertent arterial puncture and cardiac tamponade as complications associated only with non-tunnelled CVCs, but not tunnelled CVCs. Given that the insertion technique is largely similar (except for the tunnelling step), these complications should also be considered for tunnelled catheters unless there is strong evidence suggesting otherwise.

Additionally, nerve injury is listed solely as a complication of PICCs. Although recent data may be limited, there are published case reports of nerve injuries associated with CVC insertion.

The authors should either provide justification for this classification or revise the table to reflect the possibility of such complications across access types. While the authors note that recent data were “not reported,” this should not preclude acknowledging the existence of reports or other evidence. For completeness and transparency, this type of data should be acknowledged, particularly when discussing potentially serious complications.

Author Response

Response to Reviewer 2 Comments

1. Summary

Thank you very much for taking the time to review this manuscript. Please find the detailed responses below and the corresponding corrections in track changes in the re-submitted files.

3. Point-by-point response to Comments and Suggestions for Authors

Comments 1: The authors state in Table 1 (lines 104–105) that both hydrophobic and hydrophilic polymer coatings have antithrombogenic properties. While this may be true from a theoretical or in vitro perspective, it is surprising. The conclusion that both types of coating are antithrombogenic should be further contextualized.

Response 1: Thank you for your comment. As you have pointed out, these in vitro results are surprising, given that the underlying concepts are directly conflicting. We have attempted to provide clarity for the reader by outlining the proposed mechanisms in Table 1. Most descriptions of why these catheters are classified as antithrombogenic originate from in vitro models, as referenced in the article (Table 1 references [6, 12, 14–16]).

Hydrophilic catheter coatings were first introduced in the 1980s, while hydrophobic polymers are more recent additions. The Therapeutic Goods Administration in Australia approved the BioFlo PICC in December 2012 (https://www.tga.gov.au/resources/artg/203832). However, as we have mentioned, several human clinical trials have shown that different catheter coatings do not demonstrate efficacy in reducing thrombosis or occlusion, perhaps suggesting that they are not as anti-thrombogenic as initially hoped. We have updated the text to make this point clearer.

Changes made to the text:

Lines 101-105: Table 1 describes the proposed mechanisms underlying antithrombogenic coatings, most of which originate from in vitro models. While in vitro studies of these materials have been promising, there is limited evidence of the efficacy of supposed anti-thrombogenic PICC coatings in human trials.

Comments 2: The authors describe in Table 4 (lines 178–181) that only CVCs may be inserted into femoral veins. This is an oversimplification. Tunnelled CVCs can also be placed via femoral access, particularly in patients where upper body access is contraindicated. Clarifying this point would improve the objectivity and completeness of the discussion on vascular access selection. It would also help avoid under-representing the versatility of tunnelled catheters in clinical practice.

Response 2: Thank you for the comment. You are correct, and this was an oversight on the part of the authors. We have corrected the text within Table 4 to reflect this.  

Changes made to the text:

Table 4 (Line 179-182): Column of tunnelled CVCs changed to include “Jugular, subclavian or femoral vein” and “Cavo-atrial junction (except femoral catheters)”

Comments 3: It is unclear why inadvertent arterial puncture is not listed as a potential complication of PICC insertion (Table 5). The basilic vein runs in close proximity to the brachial artery, and arterial puncture has been described in the literature as a complication of PICC insertion. This omission may create an impression of differential risk that is not fully supported by the evidence.

Similarly, the table reports inadvertent arterial puncture and cardiac tamponade as complications associated only with non-tunnelled CVCs, but not tunnelled CVCs. Given that the insertion technique is largely similar (except for the tunnelling step), these complications should also be considered for tunnelled catheters unless there is strong evidence suggesting otherwise.

Additionally, nerve injury is listed solely as a complication of PICCs. Although recent data may be limited, there are published case reports of nerve injuries associated with CVC insertion.

The authors should either provide justification for this classification or revise the table to reflect the possibility of such complications across access types. While the authors note that recent data were “not reported,” this should not preclude acknowledging the existence of reports or other evidence. For completeness and transparency, this type of data should be acknowledged, particularly when discussing potentially serious complications.

Response 3: Thank you very much for your comment. You are correct in pointing out that these are all documented complications for these procedures. We did not mean to suggest that these were  not recognised complications and apologize for the miscommunication. The table was developed to display recent prevalences of complications seen in large randomised controlled trials or meta-analyses (from the past 10 years) that were included in this literature search. Of the studies included in our review, none of these large studies reported prevalences of these complications (PICC arterial puncture and nerve injury, tunnelled catheter arterial puncture and cardiac tamponade, etc.). We have updated the wording of the table to enhance clarity, marking unavailable data with “NA” instead of “X” and providing additional information in the table description.

Changes made to the text:

Table 5 (Line 196-197): Substituted “X” for “NA”.

“Abbreviations: Data not available in reviewed articles (NA).” 

Reviewer 3 Report

Comments and Suggestions for Authors In this review, the authors provide a practical approach to managing Peripherally Inserted Central Catheter-associated complications and offer recommendations for prevention strategies. They also conduct a thorough review of the pertinent literature. I find this review comprehensive and believe it provides valuable guidance for clinicians in selecting appropriate devices for central venous access and in developing strategies to prevent and treat complications associated with PICC insertion and maintenance.    The conclusions, based on comprehensive literature analysis, are well-supported and appropriately highlight the necessity of catheter selection based on individual patient clinical characteristics and comorbidities. The bibliography demonstrates a well-balanced approach to literature selection, with more than half of the references from the last 5 years, ensuring inclusion of current evidence and recent technological advances.  However, I have several corrections and suggestions that could enhance the manuscript's quality  
Page Line Manuscript Comments
2 83 there is currently limited high-quality evidence Please clarify high quality evidence
2 85 saw a significant reduction Please clarify significant
2 87, 90   In the text, reference numbers should be placed before the punctuation. Pease put the reference in the correct place
4 139-146   Please mention the references here
 5   Figure 1 Please add the corresponding copyright-related content in the caption if required (please refer to Intellectual Property i.a. Copyright, Patent and Licensing)
5 299-300   Please mention the references here
6   Table 4 The table contains mixed bullet point styles within cells. Please choose one bullet format (e.g., solid bullets •) and apply it consistently across all table entries
10 331-333   Please mention the references here
11 360, 366, 372   The subheading notation in Chapter 7 lacks consistency. First section uses numerical formatting (7.1). Please apply a Please uniform numbering style throughout this chapter to improve document structure and readability.
11   Table 6 Consider applying a consistent table style throughout the document for better presentation
 12    Table 7 Consider applying a consistent table style throughout the document for better presentation
 20   29. Upper Limb and Thoracic Veins.; 2019. Accessed October 6, 2025. https://www.pharmacy180.com/article/veins-and-their-563 branches-3634/ The access data for web sources in references 29 are dated after the manuscript submission. Please correct these to show the actual dates when the sources were accessed during manuscript preparation
  I hope these comments will assist the authors with their revisions and help the Academic Editor make an informed decision on the manuscript. If necessary, I am available to provide additional feedback following the authors' response. I maintain my recommendation for publication with minor revisions.

Author Response

Response to Reviewer 3 Comments

1. Summary

Thank you very much for taking the time to review this manuscript. Please find the detailed responses below, along with the corresponding corrections in track changes in the resubmitted files.

3. Point-by-point response to Comments and Suggestions for Authors

Comments 1: Page 2 Line 83 there is currently limited high-quality evidence. Please clarify high quality evidence.

Response 1: Thank you for your comment. We have modified the text below to improve the clarity of this statement. 

Line 83-84: “However, there is currently mixed evidence regarding the effectiveness of antimicrobial coating on PICC lines”

Comments 2: Page 2 Line 85 saw a significant reduction. Please clarify significant

Response 2: Thank you for bringing this to our attention. We have included the statistical significance with this statement.

Line 86 “A 2016 meta-analysis of 8 studies, including 12,879 patients, found a significant reduction in CLABSI with antimicrobial-coated PICCS (relative risk 0.29; 95% CI, 0.10-0.78), the effect of which was more pronounced in patient populations at greater baseline risk of infection [8].”

Comments 3: Page 2 Line 87, 90 In the text, reference numbers should be placed before the punctuation. Pease put the reference in the correct place

Response 3: Thank you for bringing this to our attention, We have corrected this punctuation error.

Line 89- 91: “infection [8]. However, a larger subsequent meta-analysis of 12 studies, with a total of 51,373 patients, failed to replicate this result, demonstrating no statistically significant reduction in CLABSI with antimicrobial-coated lines [9].”

Comments 4: Page 4 Line 139-146 Please mention the references here

Response 4: Thank you for your comment. We have included the appropriate references to improve accuracy within the text.

Line 142-150: These include the administration of irritant medications, such as chemotherapy and vasoactive agents, as well as prolonged courses of intravenous medications, including antimicrobials [26,27]. PICCs are also commonly used to infuse hyperosmolar solutions or substances with extreme pH, such as TPN. Patients may also benefit from PICC placement in cases that require frequent venous access, such as those involving multiple transfusions, poor peripheral access, or repeated blood sampling. Other indications for PICC insertion include anatomic abnormalities in the chest and neck that make standard CVC placement difficult, as well as when required as an alternative to umbilical catheters in neonates [26,27].

Comments 5:  Figure 1    Please add the corresponding copyright-related content in the caption if required (please refer to Intellectual Property i.a., Copyright, Patent and Licensing)

Response 5: Thank you for your comment. From our search, we could not find any relevant copyright for the image. To confirm this, I have emailed the authors of the website for permission and await their response.

Comments 6: Page 5 Line 299-300 Please mention the references here

Response 6: Thank you for the comment. We have included references to support the relevant statements.

Lines 306: “Catheter occlusion is a significant failure, with potential risk of life-threatening loss of venous access in critical care settings. Secondary effects may include economic costs and patient distress from catheter re-insertion [26, 46].”

Comments 7: Table 4. The table contains mixed bullet point styles within cells. Please choose one bullet format (e.g., solid bullets •) and apply it consistently across all table entries.

Response 7: Thank you. We have modified the text to make the table more consistent, using a single solid bullet style.

Table 4 (page 6): "Substances able to be safely administered” column has been changed to use solid bullet points

Comments 8: 331-333 Please mention the references here

Response 8: Thank you for your comment. We have corrected the references in this section to refer to the appropriate references and our table presenting this data. 

Lines 339-342: Historically, this risk has been considered lower than that of standard CVCs [2,3]. Notably, the risk of serious complications such as large artery puncture, pneumothorax, haemothorax and cardiac tamponade, is reduced compared to traditional CVCs, as presented in Table 5.

Comments 9: 360, 366, 372 The subheading notation in Chapter 7 lacks consistency. First section uses numerical formatting (7.1). Please apply a Please uniform numbering style throughout this chapter to improve document structure and readability.

Response 9: Thank you for bringing this to our attention. You are correct, and we have updated the text to include consistent subheadings.

Line 343: 7.1 “Malpositioning”

Line 379: 7.2 “Bleeding/ haematoma formation”

Line 385: 7.3 “Nerve injury”

Line 391: 7.4 “Rare complications”

Comments 10: Table 6 and 7 Consider applying a consistent table style throughout the document for better presentation.

Response 10: Thank you for your comment. This is an excellent point, and we have modified the formatting of all tables within the text to be consistent (excluding appendix tables, which are differentiated from the body of the article). 

Table 6 and 7 on lines 366 and 473 respectively have been updated to match the formatting of the rest of the in-text tables.

Comments 11: 29. Upper Limb and Thoracic Veins.; 2019. Accessed October 6, 2025. https://www.pharmacy180.com/article/veins-and-their-563 branches-3634/ The access data for web sources in references 29 are dated after the manuscript submission. Please correct these to show the actual dates when the sources were accessed during manuscript preparation.

Response 11: Thank you for this comment. Apologies for the transcription error; this was meant to be June 6th 2025. We have modified the reference to correct this error.

Line 630: Upper Limb and Thoracic Veins.; 2019. Accessed June 6, 2025. https://www.pharmacy180.com/article/veins-and-their-branches-3634/

Reviewer 4 Report

Comments and Suggestions for Authors

I have read with interest the manuscript submitted about PICC: Peripherally Inserted Central Catheter-associated complications and recommendation.

I really liked to read the manuscript, But I am not sure it  has the merits to be published as a narrative review. I would like to ask the authors does this review adds significant contribution to the literature or it just repeats the already known stuff.

I would like to appreciate the authors for their time and effort.

Author Response

Response to Reviewer 4 Comments

Thank you for taking the time to review this manuscript. Please find our response below.

3. Point-by-point response to comments and Suggestions for Authors

Comments 1: I have read with interest the manuscript submitted about PICC: Peripherally Inserted Central Catheter-associated complications and recommendations.

I really liked reading the manuscript, but I am not sure it has the merits to be published as a narrative review. I would like to ask the authors: Does this review add significant contribution to the literature, or does it just repeat the already known information?

I would like to appreciate the authors for their time and effort.

Response 1:

We thank you kindly for taking the time to read our paper. We are sorry to hear that you do not believe it makes a significant contribution to the scientific body of knowledge. We strongly believe it has a significant contribution to literature for the following reasons.

  1. Firstly, this article provides a summary of differences between several different intravascular devices and recent evidence for complication rates between devices, something which the current literature has not commonly done. Many reviews focus on a single device or compare only two. This review presents a comprehensive comparison of complications across all devices in a single paper.
  2. Similarly, we have provided a comprehensive examination of almost all possible complications, collating information regarding their risk factors and prevention in a single article for clinicians to access.
  3. Thirdly, all the results referenced have been from the past 10 years, with the majority being from the past 5 years- thus providing a contemporary update on the topic. This is especially important as the PICC procedure and available devices have evolved over time (as described in the article), and thus the incidence rate of complications seen is not the same as they were >10years ago.
  4. Finally, we have tried to provide a well-referenced, evidence-based resource for clinicians to refer to and gather information quickly, thus allowing them to choose the most appropriate device and patient selection to implement within their practice.

Thank you again for your valuable comments.

Round 2

Reviewer 4 Report

Comments and Suggestions for Authors

I do not think any revisions are required, as any revision will not overcome the scientific contribution. I have some reservations regarding its novelty and contribution to the existing literature. The article also lacks critical analysis. I would like to also ask the proposed recommendations based on recent evidence or expert consensus that hasn’t been consolidated elsewhere

Author Response

I agree there is no revision required 
An article like ours (Perils of the PICC can be extended with many more pages - however, the manuscript is already 25 pages at the moment.
We wanted to produce a NARRATIVE review, not a systematic review and meta-analysis.
We have included 88 references, which is a significant number.
The main take-home message is that PICC lines are often considered easy catheters that do not harm the patient, and given their name, peripherally inserted catheters, it is assumed that they do not have complications. THAT IS A MISCONCEPT by many doctors. Our aim was to educate doctors (intensivists, anaesthetists, and internal medicine specialists, ...) to make them aware of the risks associated with PICC. 
We have consistently demonstrated that the incidence of complications using the PICC line is significantly higher than with a CVC (central venous catheter). 
We have made that very clear in both the abstract, the main manuscript and in the conclusions.
I hope readers will consider alternatives to PICC lines if they need to be used in an Intensive Care setting or an Operating Room. They do not have an indication there; the only indication I see for PICC lines is on the ward, for example, in oncology. In my institutions, we have almost abandoned PICC lines in the ICU and operating rooms.